# MOOSE: TARGETED SMALL MOLECULE GENERATION VIA MULTI-OBJECTIVE-GUIDED DISCRETE DIFFUSION

**Elizabeth H. Mahood**[1], **William J. Pattie**[2], **Ethan M. Jones**[2], **Sophia Vincoff**[1], **Adi Mashiach**[2],
**Pranam Chatterjee**[1,3,†]

[1]Department of Bioengineering, University of Pennsylvania
[2]Output Biosciences, USA
[3]Department of Computer and Information Science, University of Pennsylvania
[†]Corresponding author: pranam@seas.upenn.edu

## ABSTRACT

We introduce **MOOSE**, a **M**ulti-**O**bjective **O**ptimization framework for property-driven **S**mall-molecule **E**ngineering using discrete diffusion. MOOSE builds on *GenMol*, a masked discrete diffusion model that generates molecules in a fragment-based token space, and adapts it to goal-directed molecular design through iterative fine-tuning toward desired property distributions. By scoring target binding affinity with a rapid, state-of-the-art sequence-based surrogate model (Bonbon), together with standard chemical heuristics for drug-likeness and synthetic accessibility, MOOSE enables scalable optimization without reliance on docking during generation. We evaluate MOOSE on a diverse set of protein targets relevant to therapeutic discovery. Across all targets, molecules generated by MOOSE achieve stronger predicted binding affinity compared to known reference ligands, while maintaining chemical diversity, synthesizability, and drug-likeness. Together, these results demonstrate that MOOSE can be directly integrated into practical small-molecule discovery pipelines, enabling scalable multi-objective optimization without docking-in-the-loop.

## 1 INTRODUCTION

The discovery and design of small molecules remains a central challenge in drug, agrochemical, and materials science (Bedard et al., 2020). Practical molecular discovery requires navigating a vast combinatorial space while satisfying multiple competing constraints, including target binding, synthetic accessibility, drug-likeness, pharmacokinetics, and toxicity (Fromer & Coley, 2023; Brown et al., 2019). While classical pipelines rely on virtual screening and docking-based prioritization (Cosconati et al., 2010; da Rocha et al., 2025), these approaches scale poorly, motivating the integration of rapid surrogate property predictors with generative model proposers to prioritize a small number of high-confidence candidates (Swanson et al., 2024; Siramshetty et al., 2023; Fu et al., 2024; Lee et al., 2025).

Recent advances in generative modeling have enabled scalable exploration of chemical space (Du et al., 2024; Zeng et al., 2022). Most classically, structure-based and graph-based generators conditioned on fixed protein-ligand poses (Isert et al., 2023; Xu et al., 2023; Stark et al., 2025; Hoogeboom et al., 2022; Lin et al., 2024) are effective when high-quality structures are available, but are limited otherwise. In parallel, autoregressive *sequence* models over SMILES representations (Li et al., 2023; Wang et al.) have been developed to improve scalability to diverse targets, but struggle to enforce global structural constraints such as ring closures and fragment compatibility (Chen & Jung, 2024). These limitations have driven interest in discrete, non-autoregressive generative models. Most notably, discrete diffusion and discrete flow matching have become the dominant formulations, as they frame the generative process as transport over sequence space, enabling parallel refinement of entire molecules (Austin et al., 2021; Sahoo et al., 2024; Shi et al., 2024; Gat et al., 2024; Stark et al., 2024; Tang et al., 2025b; Davis et al., 2024). Multi-objective optimization algorithms have been successfully developed for discrete generative models, specifically for peptides and nucleic acids (Tang et al., 2025a; Chen et al., 2025b;a), but small-molecule diffusion models have thus far

focused primarily on feasibility constraints (Cardei et al.), leaving multi-objective, target-specific optimization largely unexplored.

In this setting, the throughput of binding-affinity prediction becomes one of the primary limitations. Docking pipelines such as AutoDock Vina (Eberhardt et al., 2021) and structure-based predictors such as Boltz-2 (Passaro et al., 2025) help to provide high-quality estimates but are *far too slow* at scale: Boltz-2 typically requires 20-30 seconds per molecule, which (in the settings of this paper) would translate to roughly 60-90 days of computation to complete fine-tuning across 13 targets. While faster retrieval-based approaches such as DrugCLIP (Jia et al., 2026) can precompute target-drug similarity scores over embeddings on fixed molecular libraries, they do not provide meaningful affinity predictions and are unsuited to generative optimization, which requires on-the-fly evaluation.

To address the lack of multi-objective, target-specific optimization for small-molecule generative models, we introduce a transport-driven framework that extends discrete diffusion to property-guided small-molecule engineering. Our **contributions** are as follows:

1. We introduce **MOOSE**, a **M**ulti-**O**bjective **O**ptimization framework for **S**mall-molecule **E**ngineering that performs distribution-level generation and optimization via transport-driven fine-tuning of discrete diffusion models.

2. We build MOOSE on *GenMol*, a fragment-based masked discrete diffusion generator over SAFE molecular sequences (Lee et al., 2025), and adapt it to goal-directed design using TR2-D2, which reshapes the generator distribution through trajectory-aware fine-tuning (Tang et al., 2025c).

3. We enable scalable optimization by incorporating a state-of-the-art high-throughput surrogate predictor, **Bonbon**, which can process over 2,000 molecules per second on a single H100 GPU.

4. We evaluate MOOSE under a practical top-$k$ discovery protocol and demonstrate consistent improvements over known reference ligands across diverse targets, achieving favorable trade-offs between predicted binding affinity, drug-likeness (QED), and synthetic accessibility (SA).

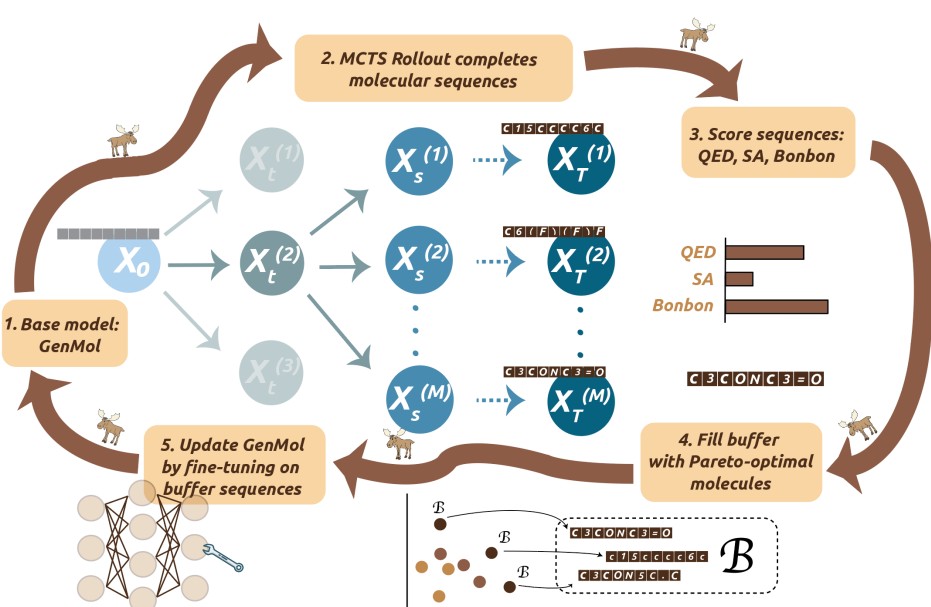

Figure 1: Overview of **MOOSE**. GenMol serves as the base masked discrete diffusion generator. Tree-search-guided rollouts complete molecular sequences, which are scored using multiple surrogate objectives (QED, SA, Bonbon). Pareto-optimal sequences populate a replay buffer, and the generator is updated via transport-driven fine-tuning following the TR2-D2 framework.

## 2 METHODS

### 2.1 BASE GENERATOR: GENMOL

Let $p_{\theta_0}(x)$ denote a pre-trained masked discrete diffusion language model defining an unconditional distribution over molecular sequences $x$. We instantiate $p_{\theta_0}$ using GenMol (Lee et al., 2025), which operates over SAFE molecular representations Noutahi et al. (2024) and generates molecules via non-autoregressive, bidirectional denoising. GenMol provides a fragment-level prior over chemically valid small molecules and serves as the base generator for optimization.

### 2.2 OBJECTIVE-GUIDED DISTRIBUTIONAL TARGET

Given an external objective $S(x)$, such as predicted binding affinity or developability properties, the goal of molecular optimization is to bias $p_{\theta_0}(x)$ toward high-scoring molecules while preserving its learned prior. Following TR2-D2 (Tang et al., 2025c), this is formalized via the reward-tilted target distribution

$$p^*(x) \propto p_{\theta_0}(x) \exp\left(\frac{S(x)}{\alpha}\right), \tag{1}$$

where $\alpha > 0$ controls the strength of deviation from the base model. This defines an implicit energy-based reweighting without modifying the diffusion corruption process.

### 2.3 AMORTIZED FINE-TUNING WITH WDCE

Direct sampling from $p^*(x)$ is intractable. TR2-D2 therefore learns a new parameterization $p_\theta(x)$ that approximates $p^*(x)$ via amortized fine-tuning. Let $\boldsymbol{X}_{0:T}$ denote a diffusion trajectory with terminal sequence $\boldsymbol{x} = \boldsymbol{X}_T$. Fine-tuning minimizes the weighted denoising cross-entropy

$$\mathcal{L}_{\text{WDCE}}(\theta) = \mathbb{E}_{\boldsymbol{X}_{0:T} \sim \mathbb{P}^v}\left[w(\boldsymbol{X}_{0:T})\mathcal{L}_{\text{DCE}}(\theta; \boldsymbol{X}_T)\right], \tag{2}$$

where $\mathbb{P}^v$ is a proposal trajectory distribution and the importance weight is

$$w(\boldsymbol{X}_{0:T}) \propto \exp\left(\frac{S(\boldsymbol{X}_T)}{\alpha}\right) \prod_t \frac{p_{\theta_0}(\boldsymbol{X}_{t-1} \mid \boldsymbol{X}_t)}{p_{\bar{\theta}}(\boldsymbol{X}_{t-1} \mid \boldsymbol{X}_t)}. \tag{3}$$

Fine-tuning updates only the denoising conditionals, leaving the corruption schedule unchanged.

Fine-tuning was performed on NVIDIA H100 GPUs. A full 200-epoch TR2-D2 fine-tuning run requires approximately 40 minutes on a single H100 GPU and generates approximately 21,000 molecules. Bonbon evaluation contributes under 11 seconds to this runtime. Substituting structure-based predictors such as Boltz-2 (Passaro et al., 2025) would increase runtime to multiple days per target, making iterative multi-target optimization infeasible.

### 2.4 AFFINITY ORACLE: BONBON

Bonbon is a Transformer-based model for protein-small molecule binding and affinity prediction developed by Output Biosciences. Briefly, the model contains approximately 800 million parameters and was trained using a self-supervised learning objective designed for the relational structure of molecular interaction data. Training data comprised of paired protein and small-molecule sequences from Output Biosciences' proprietary high-throughput screening datasets, enriched with synthetic interaction data generated from experimentally measured binding distributions. In total, Bonbon was trained on approximately 3.2 trillion tokens, over two orders of magnitude larger than publicly available binding datasets such as BindingDB (Liu et al., 2025). Training was conducted on a cluster of NVIDIA H100 GPUs for 6.5 days using the AdamW optimizer with learning rate $1 \times 10^{-3}$, 10,000 warmup steps, and full sharded data parallelism.

Bonbon achieves state-of-the-art performance on binding and affinity prediction while running at 2,000 interactions per second, orders of magnitude faster than structure-based methods. On the MF-PCBA benchmark simulating high-throughput screening (Buterez et al., 2023), Bonbon achieves AUPR of 0.056 compared to Boltz-2 AUPR of 0.025 and Boltz-2 ipTM of 0.005. On binding affinity prediction, Bonbon achieves per-protein Pearson correlation of 0.692 on FEP+4 (Boltz-2: 0.66) and 0.687 on OpenFE (Boltz-2: 0.62). Bonbon also outperforms sequence-based retrieval models: on the LIT-PCBA benchmark, Bonbon achieves BEDROC of 11.9% and EF 1% of 9.79 (DrugCLIP: 6.23% and 5.5, respectively).

Bonbon produces a scalar affinity score for each protein-molecule pair and was used as the primary binding-affinity objective within MOOSE. During optimization, Bonbon was treated as a fixed oracle and was not updated.

## 2.5 Target Selection

We considered only targets with PDB co-crystal structures containing a known reference binder, and excluded any target whose reference binder did not outscore random molecules in Vina docking.

## 2.6 Molecular docking

Docking was used only for post-hoc validation. Molecular docking was performed using AutoDock Vina (Eberhardt et al., 2021). For each target, a PDB structure containing a co-crystallized reference binder (i.e. known drug) was selected (Table S1). Receptors were prepared by removing solvent and ligands, adding hydrogens using ChimeraX (Pettersen et al., 2021), and converting to PDBQT format using Meeko (Santos-Martins et al., 2025). Generated SMILES were converted to single conformers using RDKit ETKDG (Wang et al., 2020), energy minimized with MMFF94s, and docked with exhaustiveness 16. The top-scoring pose was reported.

## 2.7 *In silico* evaluation

Following the Practical Molecular Optimization benchmark (Gao et al., 2022), we evaluate the top-10 molecules ranked by Bonbon score for each target. For each molecule, we compute synthetic accessibility, quantitative estimate of drug-likeness (Bickerton et al., 2012), and Vina docking scores. Statistical significance is assessed using one-sided Mann-Whitney U tests comparing GenMol and MOOSE.

## 3 Results

We first assessed whether MOOSE can recover or improve upon known active compounds for each target. Across a diverse panel of kinases, proteases, and metabolic enzymes, the top molecules generated by MOOSE achieve docking scores that are comparable to or better than those of established reference ligands, while simultaneously exhibiting higher QED and lower synthetic accessibility scores (Table 1). This combination indicates a favorable trade-off between predicted potency and developability, suggesting that optimization is not achieved at the expense of chemical practicality; representative docking poses for two targets are shown in Figure 2.

Table 1: Top **MOOSE**-generated molecule vs. known active compound for each target. Vina scores in kcal/mol; best values per target in bold.

| Target Protein | UniProt | Type | VINA (kcal/mol; ↓) | QED (↑) | SA (↓) |
|---|---|---|---|---|---|
| Mitogen-activated protein kinase 10 (JNK3) | P53779 | Tanzisertib | −10.218 | 0.548 | 3.455 |
| | | **MOOSE** | **−10.517** | **0.706** | **2.141** |
| Protein-tyrosine kinase 2-beta (FAK2) | Q14289 | Defactinib | **−8.763** | 0.416 | **2.829** |
| | | **MOOSE** | −8.717 | **0.593** | 3.669 |
| Ca/calmodulin-dep. protein kinase II $\delta$ (CAMK2D) | Q13557 | CHEMBL4215541 | −8.569 | 0.547 | **2.411** |
| | | **MOOSE** | **−9.398** | **0.587** | 2.661 |
| Apoptosis regulator (BCL2) | P10415 | Navitoclax | **−10.719** | 0.105 | 4.131 |
| | | **MOOSE** | −9.758 | **0.665** | **2.201** |
| Histamine N-methyltransferase (HNMT) | P50135 | Diphenhydramine | −8.235 | **0.785** | **1.815** |
| | | **MOOSE** | **−10.873** | 0.384 | 2.437 |
| Casein kinase I isoform epsilon (CSNK1E) | P49674 | Umbralisib | −9.232 | 0.238 | 3.467 |
| | | **MOOSE** | **−9.448** | **0.842** | **2.002** |
| Collagenase 3 (MMP-13) | P45452 | Rebimastat | −7.207 | 0.265 | 3.947 |
| | | **MOOSE** | **−11.643** | **0.295** | **2.543** |
| Ephrin type-B receptor 4 (EPHB4) | P54760 | Dasatinib | −7.967 | 0.466 | 2.650 |
| | | **MOOSE** | **−9.466** | **0.574** | **2.355** |
| Serine/threonine-protein kinase (PLK1) | P53350 | Adavosertib | −9.953 | 0.374 | 3.001 |
| | | **MOOSE** | **−11.277** | **0.594** | **2.273** |
| NT-3 growth factor receptor (NTRK3) | Q16288 | Entrectinib | −10.927 | 0.294 | 2.845 |
| | | **MOOSE** | **−11.847** | **0.463** | **2.134** |
| 4-hydroxyphenylpyruvate dioxygenase (HPD) | P93836 | Nitisinone | −9.126 | 0.295 | 2.546 |
| | | **MOOSE** | **−10.485** | **0.656** | **2.479** |
| Plasmepsin II (PMII) | P46925 | CHEMBL3763404 | −9.131 | 0.548 | 3.035 |
| | | **MOOSE** | **−10.281** | **0.643** | **3.020** |
| Bifunctional DHFR-thymidylate synthase (DHFR-TS) | Q07422 | Pyrimethamine | −7.722 | **0.856** | **2.123** |
| | | **MOOSE** | **−9.268** | 0.664 | 2.697 |

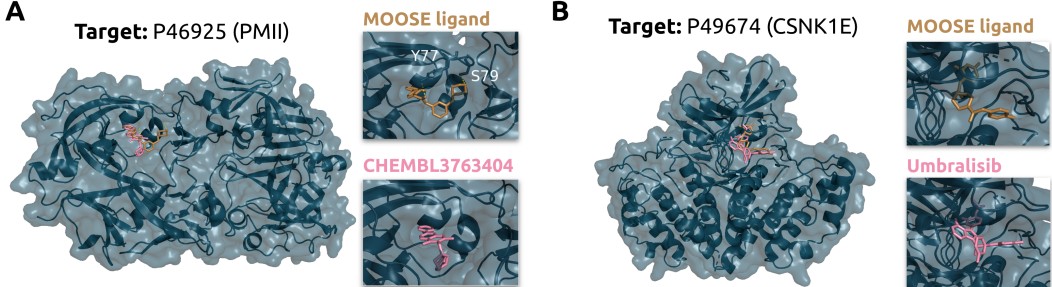

Figure 2: Post-hoc docking and structural visualization of representative MOOSE-generated ligands. Top-ranked MOOSE molecules were docked into Plasmepsin II (PMII) and Casein kinase I epsilon (CSNK1E) using AutoDock Vina and visualized in PyMOL, alongside known reference ligands.

To quantify the effect of transport-driven fine-tuning relative to the base generator, we next compared MOOSE directly against GenMol using AutoDock Vina (Lee et al., 2025; Eberhardt et al., 2021). Across all evaluated targets (Table S1), MOOSE produces substantially lower docking energies, with improvements that are statistically significant under one-sided Mann-Whitney U tests (Table S2). The largest gains are observed for targets such as JNK3, PLK1, and Ephrin-B4, highlighting the ability of fine-tuning to consistently shift the generator toward higher-affinity regions of chemical space.

Beyond docking, MOOSE also improves surrogate-based measures of binding and developability. Across all targets, predicted binding affinity as measured by Bonbon increases under fine-tuning, while QED is maintained or improved and synthetic accessibility improves consistently (Table S3). This pattern indicates that MOOSE performs genuine multi-objective optimization rather than overfitting to a single scoring function, preserving chemical plausibility while improving predicted activity.

Taken together, the use of Bonbon for binding affinity prediction alongside lightweight chemical heuristics enables efficient large-scale optimization in practice. A complete fine-tuning run completes in under one hour on a single H100 GPU, whereas replacing these surrogates with structure-based predictors such as Boltz-2 (Passaro et al., 2025) would increase runtime to several days per target, rendering multi-target optimization infeasible. This efficiency is critical for applying transport-driven generative optimization in realistic discovery settings.

## 4 DISCUSSION

In this study, we introduce **MOOSE**, a transport-based framework for multi-objective small-molecule optimization that integrates GenMol (Lee et al., 2025), TR2-D2 (Tang et al., 2025c), and a state-of-the-art high-throughput surrogate predictor, **Bonbon**, for binding affinity, together with standard chemical heuristics for drug-likeness and synthetic accessibility. Across diverse targets, MOOSE generates molecules with improved predicted binding affinity and docking scores relative to both known reference ligands and the base GenMol generator, while maintaining chemical diversity and developability. These results indicate that distribution-level optimization of discrete diffusion models, enabled by fast binding-affinity surrogates and lightweight chemical objectives, offers a practical alternative to docking-centric screening pipelines for early-stage discovery.

A primary limitation of this study is that all evaluations are based on surrogate models and *in silico* metrics, without prospective experimental validation. In addition, the quality of optimization is bounded by the fidelity of the surrogate objectives and the choice of fine-tuning hyperparameters. Future work will focus on wet-lab validation of MOOSE-generated candidates and extending the framework to incorporate additional objectives, including selectivity, toxicity, and broader ADME-related properties, as reliable high-throughput surrogates for these endpoints become available. Developing standardized benchmarks for distribution-level molecular optimization remains an important direction for the field.

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

## A   APPENDIX

Table S1: Protein targets and structural data used for molecular docking. PDB structures were selected based on availability of co-crystallized ligands, which define the docking box center.

| Protein | UniProt | PDB | Ligand ID |
|---|---|---|---|
| Mitogen-activated protein kinase 10 (JNK3) | P53779 | 3TTI | KBI |
| Protein-tyrosine kinase 2-beta (FAK2) | Q14289 | 3FZR | 3JZ |
| Ca/calmodulin-dep. protein kinase II $\delta$ (CAMK2D) | Q13557 | 6AYW | C2V |
| Apoptosis regulator (BCL2) | P10415 | 4LVT | 1XJ |
| Histamine N-methyltransferase (HNMT) | P50135 | 2AOT | SAH |
| Casein kinase I isoform epsilon (CSNK1E) | P49674 | 4HNI | 16W |
| Collagenase 3 (MMP-13) | P45452 | 5UWN | 8O7 |
| Ephrin type-B receptor 4 (EPHB4) | P54760 | 2X9F | X9F |
| Serine/threonine-protein kinase (PLK1) | P53350 | 9D0P | 8X7 |
| NT-3 growth factor receptor (NTRK3) | Q16288 | 4YMJ | 4EJ |
| 4-hydroxyphenylpyruvate dioxygenase (HPD) | P93836 | 6J63 | NTD |
| Plasmepsin II (PMII) | P46925 | 4Z22 | 4KG |
| Bifunctional DHFR-thymidylate synthase (DHFR-TS) | Q07422 | 6AOG | CP6 |

Table S2: Vina docking scores (kcal/mol) for *GenMol* vs. **MOOSE**. $p$-values from one-sided Mann-Whitney U test.

| Target Protein | UniProt | Method | VINA (kcal/mol; ↓) | $p$-value |
|---|---|---|---|---|
| Mitogen-activated protein kinase 10 (JNK3) | P53779 | *GenMol* 
 **MOOSE** | $-7.852 \pm 0.898$ 
 $-\mathbf{9.450} \pm 0.428$ | $9.13 \times 10^{-5}$ |
| Protein-tyrosine kinase 2-beta (FAK2) | Q14289 | *GenMol* 
 **MOOSE** | $-7.513 \pm 0.819$ 
 $-\mathbf{8.183} \pm 0.315$ | $2.70 \times 10^{-2}$ |
| Ca/calmodulin-dependent protein kinase II $\delta$ (CAMK2D) | Q13557 | *GenMol* 
 **MOOSE** | $-7.629 \pm 0.856$ 
 $-\mathbf{8.341} \pm 0.603$ | $3.78 \times 10^{-2}$ |
| Apoptosis regulator (BCL2) | P10415 | *GenMol* 
 **MOOSE** | $-7.373 \pm 0.903$ 
 $-\mathbf{8.455} \pm 0.977$ | $1.29 \times 10^{-2}$ |
| Histamine N-methyltransferase (HNMT) | P50135 | *GenMol* 
 **MOOSE** | $-7.644 \pm 0.840$ 
 $-\mathbf{8.833} \pm 1.367$ | $1.88 \times 10^{-2}$ |
| Casein kinase I isoform epsilon (CSNK1E) | P49674 | *GenMol* 
 **MOOSE** | $-7.536 \pm 0.804$ 
 $-\mathbf{8.632} \pm 0.667$ | $3.64 \times 10^{-3}$ |
| Collagenase 3 (MMP-13) | P45452 | *GenMol* 
 **MOOSE** | $-9.474 \pm 0.983$ 
 $-\mathbf{10.631} \pm 0.884$ | $7.01 \times 10^{-3}$ |
| Ephrin type-B receptor 4 (EPHB4) | P54760 | *GenMol* 
 **MOOSE** | $-7.383 \pm 0.600$ 
 $-\mathbf{8.871} \pm 0.605$ | $6.57 \times 10^{-4}$ |
| Serine/threonine-protein kinase (PLK1) | P53350 | *GenMol* 
 **MOOSE** | $-7.820 \pm 1.054$ 
 $-\mathbf{10.550} \pm 0.507$ | $9.13 \times 10^{-5}$ |
| NT-3 growth factor receptor (NTRK3) | Q16288 | *GenMol* 
 **MOOSE** | $-8.312 \pm 0.928$ 
 $-\mathbf{9.903} \pm 0.914$ | $1.10 \times 10^{-3}$ |
| 4-hydroxyphenylpyruvate dioxygenase (HPD) | P93836 | *GenMol* 
 **MOOSE** | $-7.662 \pm 1.101$ 
 $-\mathbf{9.293} \pm 0.732$ | $1.81 \times 10^{-3}$ |
| Plasmepsin II (PMII) | P46925 | *GenMol* 
 **MOOSE** | $-7.738 \pm 1.042$ 
 $-\mathbf{9.144} \pm 0.599$ | $3.64 \times 10^{-3}$ |
| Bifunctional DHFR-thymidylate synthase (DHFR-TS) | Q07422 | *GenMol* 
 **MOOSE** | $-7.803 \pm 0.857$ 
 $-\mathbf{8.460} \pm 0.450$ | $1.88 \times 10^{-2}$ |

Table S3: **Bonbon** scores, QED, and synthetic accessibility (SA) for *GenMol* vs. **MOOSE**. Best values per target in bold.

| Target Protein | UniProt | Method | Bonbon ($\uparrow$) | QED ($\uparrow$) | SA ($\downarrow$) |
|---|---|---|---|---|---|
| Mitogen-activated protein kinase 10 (JNK3) | P53779 | *GenMol* | $6.267 \pm 0.231$ | $\mathbf{0.774} \pm 0.142$ | $3.272 \pm 0.840$ |
| | | **MOOSE** | $\mathbf{7.001} \pm 0.308$ | $0.611 \pm 0.144$ | $\mathbf{2.104} \pm 0.456$ |
| Protein-tyrosine kinase 2-beta (FAK2) | Q14289 | *GenMol* | $6.120 \pm 0.574$ | $\mathbf{0.776} \pm 0.139$ | $3.316 \pm 0.830$ |
| | | **MOOSE** | $\mathbf{6.796} \pm 0.516$ | $0.656 \pm 0.103$ | $\mathbf{2.741} \pm 0.379$ |
| Ca/calmodulin-dep. protein kinase II $\delta$ (CAMK2D) | Q13557 | *GenMol* | $6.351 \pm 0.416$ | $\mathbf{0.789} \pm 0.136$ | $3.448 \pm 0.868$ |
| | | **MOOSE** | $\mathbf{7.289} \pm 0.323$ | $0.766 \pm 0.113$ | $\mathbf{2.777} \pm 0.576$ |
| Apoptosis regulator (BCL2) | P10415 | *GenMol* | $6.068 \pm 0.358$ | $0.779 \pm 0.142$ | $3.298 \pm 0.847$ |
| | | **MOOSE** | $\mathbf{6.543} \pm 0.279$ | $\mathbf{0.817} \pm 0.096$ | $\mathbf{2.694} \pm 0.637$ |
| Histamine N-methyltransferase (HNMT) | P50135 | *GenMol* | $6.903 \pm 0.261$ | $\mathbf{0.750} \pm 0.139$ | $3.396 \pm 0.805$ |
| | | **MOOSE** | $\mathbf{7.448} \pm 0.320$ | $0.652 \pm 0.136$ | $\mathbf{2.409} \pm 0.254$ |
| Casein kinase I isoform epsilon (CSNK1E) | P49674 | *GenMol* | $7.373 \pm 0.213$ | $0.804 \pm 0.142$ | $3.098 \pm 0.944$ |
| | | **MOOSE** | $\mathbf{7.788} \pm 0.173$ | $\mathbf{0.810} \pm 0.076$ | $\mathbf{2.401} \pm 0.634$ |
| Collagenase 3 (MMP-13) | P45452 | *GenMol* | $6.388 \pm 0.163$ | $\mathbf{0.831} \pm 0.102$ | $2.643 \pm 0.691$ |
| | | **MOOSE** | $\mathbf{8.239} \pm 0.430$ | $0.551 \pm 0.096$ | $\mathbf{2.482} \pm 0.271$ |
| Ephrin type-B receptor 4 (EPHB4) | P54760 | *GenMol* | $5.955 \pm 0.314$ | $\mathbf{0.788} \pm 0.132$ | $3.787 \pm 0.602$ |
| | | **MOOSE** | $\mathbf{7.672} \pm 0.187$ | $0.672 \pm 0.090$ | $\mathbf{2.340} \pm 0.205$ |
| Serine/threonine-protein kinase (PLK1) | P53350 | *GenMol* | $5.989 \pm 0.525$ | $0.773 \pm 0.141$ | $3.331 \pm 0.906$ |
| | | **MOOSE** | $\mathbf{6.871} \pm 0.421$ | $\mathbf{0.786} \pm 0.094$ | $\mathbf{2.639} \pm 0.610$ |
| NT-3 growth factor receptor (NTRK3) | Q16288 | *GenMol* | $6.386 \pm 0.392$ | $\mathbf{0.776} \pm 0.142$ | $3.706 \pm 0.572$ |
| | | **MOOSE** | $\mathbf{7.887} \pm 0.332$ | $0.523 \pm 0.081$ | $\mathbf{2.442} \pm 0.249$ |
| 4-hydroxyphenylpyruvate dioxygenase (HPD) | P93836 | *GenMol* | $7.342 \pm 0.289$ | $\mathbf{0.789} \pm 0.132$ | $3.493 \pm 0.823$ |
| | | **MOOSE** | $\mathbf{8.873} \pm 0.040$ | $0.712 \pm 0.104$ | $\mathbf{2.258} \pm 0.129$ |
| Plasmepsin II (PMII) | P46925 | *GenMol* | $6.817 \pm 0.298$ | $\mathbf{0.770} \pm 0.146$ | $3.395 \pm 0.854$ |
| | | **MOOSE** | $\mathbf{8.061} \pm 0.337$ | $0.647 \pm 0.134$ | $\mathbf{2.175} \pm 0.425$ |
| Bifunctional DHFR-thymidylate synthase (DHFR-TS) | Q07422 | *GenMol* | $6.927 \pm 0.397$ | $\mathbf{0.773} \pm 0.136$ | $3.383 \pm 0.804$ |
| | | **MOOSE** | $\mathbf{8.804} \pm 0.233$ | $0.748 \pm 0.055$ | $\mathbf{2.915} \pm 0.209$ |

Table S4: **Default hyperparameters for MOOSE**. Please see the TR2-D2 publication for intuition on parameter functionality. $M$ = Num children; $R$ = Num WDCE replicates; $B$ = Buffer size; $c$ = Exploration Constant.

| $M$ | Seq length | $R$ | $B$ | $N_{\text{steps}}$ | $N_{\text{iter}}$ | $N_{\text{reset}}$ | $N_{\text{resample}}$ | $N_{\text{epochs}}$ | $\alpha$ | $c$ |
|---|---|---|---|---|---|---|---|---|---|---|
| 50 | 60 | 16 | 20 | 128 | 10 | 1 | 10 | 100 | 0.1 | 0.1 |

