# OpenReview forum: "MOOSE: Targeted Small Molecule Generation via Multi-Objective-Guided Discrete Diffusion"
_ICLR.cc/2026/Workshop/FM4Science — ICLR 2026 Workshop FM4Science Poster_

### Official Review · Reviewer_dS9w · 2026-02-20
**Generative active learning for target-based drug design**

**Rating:** 6
**Confidence:** 4

**Review:**

**Summary:**
The authors introduce MOOSE, a framework for active learning-based generation of protein-drug interactions. Empirical results suggest that MOOSE successfully improves the protein binding affinity rate of its generated drugs over a generative baseline (GenMol) as well as existing, well-studied drugs.

**Points of Strength:**
1. The authors demonstrate the efficacy of their proposed method by iteratively finetuning it using its highest-affinity designs, ultimately showing that evaluation metrics improve as a result.
2. The authors provide evidence of the importance of prioritizing fast, sequence-based property prediction methods in real-world drug design pipelines, to enable scalable active learning workflows.

**Points for Improvement:**
1. The authors don't describe hyperparameter tuning of MOOSE during finetuning. Tuning during finetuning could improve performance further.
2. VINA docking scores are notoriously misleading in molecular machine learning benchmarks. One remedy for interpreting them more accurately is to run an ensemble of different docking methods and to report the individual and average affinity of all methods (as separate metrics).

---

### Official Review · Reviewer_TcLm · 2026-02-22
**An efficient framework but rely on a proprietary surrogate model and lack of independent or experimental validation**

**Rating:** 9
**Confidence:** 4

**Review:**

This paper presents MOOSE, a framework for multi-objective small-molecule optimization built on GenMol and TR2-D2, using a high-throughput surrogate model Bonbon to guide binding affinity optimization. The approach integrates affinity prediction with drug-likeness (QED) and synthetic accessibility (SA) constraints, demonstrates consistent improvements over the GenMol, and achieves docking scores and surrogate affinity metrics that are comparable to, and in some cases better than, known reference ligands. The framework is computationally efficient, with fine-tuning completed in under an hour on a single GPU, making it practically appealing for early-stage drug discovery.
As acknowledged in the discussion, the study relies heavily on a proprietary surrogate model for both optimization and evaluation, which limits reproducibility and makes it difficult to assess whether the reported improvements generalize beyond the guiding model. In addition, all results are in silico, with no experimental validation, and docking is used only as a secondary surrogate rather than an independent confirmation of binding improvement. While the engineering integration is solid, wet-lab confirmation and evaluation with independent affinity predictors would substantially strengthen the scientific impact.

---

### Official Review · Reviewer_qx1Y · 2026-02-23
**Review on MOOSE**

**Rating:** 8
**Confidence:** 3

**Review:**

The authors present MOOSE, a framework designed for the multi-objective optimization of small molecules using discrete diffusion. Recognizing the scalability limitations of traditional docking-in-the-loop generative models, MOOSE builds upon the GenMol base generator and adapts it using TR2-D2, a trajectory-aware fine-tuning method. To overcome the bottleneck of slow affinity evaluations, the framework incorporates Bonbon, a high-throughput sequence-based surrogate model. The method is evaluated on a diverse panel of 13 protein targets, demonstrating that MOOSE-generated molecules achieve stronger predicted binding affinities, higher Quantitative Estimate of Drug-likeness (QED), and better Synthetic Accessibility (SA) scores compared to both the base model and known reference ligands.

The integration of the Bonbon surrogate model directly addresses the computational limitations of structure-based predictors like Boltz-2 and traditional docking pipelines like AutoDock Vina. By achieving processing speeds of 2,000 molecules per second, the pipeline is highly scalable. MOOSE-generated molecules not only outperformed the base GenMol model with statistical significance but also established favorable trade-offs between predicted potency and chemical developability.

However, there are several concerns as well. The evaluation is entirely computational, relying heavily on the accuracy of the Bonbon surrogate and post-hoc Vina docking. The lack of prospective wet-lab validation limits claims about real-world efficacy. Besides, GenMol operates over SAFE molecular representations. The performance of MOOSE is inherently bounded by the fragment-level prior and chemical space covered by this base model.

This paper presents a scalable, and computationally efficient framework for small-molecule engineering. By replacing slow docking steps with a robust surrogate model during the iterative fine-tuning of a discrete diffusion model, the authors offer a compelling tool for early-stage molecular discovery. The work is technically sound and clearly written.

---

### Meta-Review · Area_Chair_UDDB · 2026-02-28

**Recommendation:** Accept (Poster)
**Confidence:** 4

**Metareview:**

This work introduces MOOSE, a framework for multi-objective small-molecule optimization built on GenMol and and iterative fine-tuning via TR2-D2. MOOSE uses a high-throughput surrogate model Bonbon to guide binding affinity optimization. Reviewers viewed the work favorably, highlighting the performance improvements over the base GenMol generative model and the scalability of the pipeline. The fine-tuning approach appears to be a timely and relevant contribution of interest to the workshop. The primary concern shared by reviewers was the lack of wet-lab confirmation, which the authors are encouraged to pursue in future work. It is suggested as well to evaluate with independent affinity predictors to further validate the performance.

---

### Decision · Program_Chairs · 2026-03-03

Accept (Poster)